# Synthesis and Antimicrobial Activity Evaluation of Homodrimane Sesquiterpenoids with a Benzimidazole Unit

**DOI:** 10.3390/molecules28030933

**Published:** 2023-01-17

**Authors:** Lidia Lungu, Svetlana Blaja, Caleria Cucicova, Alexandru Ciocarlan, Alic Barba, Veaceslav Kulcițki, Sergiu Shova, Nicoleta Vornicu, Elisabeta-Irina Geana, Ionel I. Mangalagiu, Aculina Aricu

**Affiliations:** 1Chemistry of Natural and Biologically Active Compounds Laboratory, Institute of Chemistry, 3 Academiei Str., 2028 Chisinau, Moldova; 2“Petru Poni” Institute of Macromolecular Chemistry, Aleea Grigore Ghica Voda 41-A, 700487 Iasi, Romania; 3Metropolitan Center of Research T.A.B.O.R., 9 Closca Str., 700066 Iasi, Romania; 4Department of Research and Development, National Research and Development Institute for Cryogenics and Isotopic Technologies—ICSI Rm., Valcea, 4th Uzinei Str., 240050 Râmnicu Vâlcea, Romania; 5Faculty of Chemistry, “Alexandru Ioan Cuza” University of Iasi, 11 Carol Bd., 700506 Iasi, Romania

**Keywords:** homodrimane sesquiterpenoids, 1,3-benzimidazole unit, antifungal and antibacterial activity

## Abstract

Herein we report a feasible study concerning the synthesis and the in vitro antimicrobial activity of some new homodrimane sesquiterpenoids with a benzimidazole unit. Based on some homodrimane carboxylic acids, on their acyl chlorides and intermediate monoamides, a series of seven *N*-homodrimenoyl-2-amino-1,3-benzimidazoles and 2-homodrimenyl-1,3-benzimidazoles was synthesized. The syntheses involved the decarboxylative cyclization and condensation of the said acids or acyl chlorides with *o*-phenylendiamine and 2-aminobenzimidazole, as well as the *p*-TsOH-mediated cyclodehydration of the said monoacylamides. The structures of the synthesized compounds have been fully confirmed, including by the X-ray diffraction. Their biological activities were evaluated on five species of fungi (*Aspergillus niger*, *Fusarium solani*, *Penicillium chrysogenum*, *P. frequentans*, and *Alternaria alternata*) and two strains of bacteria (*Bacillus* sp. and *Pseudomonas aeruginosa*). Compounds **7** and **20** showed higher antifungal (MIC = 0.064 and 0.05 μg/mL) and antibacterial (MIC = 0.05 and 0.032 μg/mL) activities compared to those of the standards: caspofungin (MIC = 0.32 μg/mL) and kanamycin (MIC = 2.0 μg/mL), and compounds **4**, **10**, **14**, and **19** had moderate activities.

## 1. Introduction

In recent years, many countries have encountered microbial infections that are rapidly spreading, as they are becoming one of the most serious problems [1,2]. Those global trends stimulate the design of new molecular structures with antimicrobial properties, which could lead to new and effective medicinal preparations. Natural products have proven to be an important source of novel biologically active compounds because their natural origin implies biocompatibility, selective biological activity, and low toxicity. Drimane sesquiterpenoids are just such natural or synthetic compounds with wide spectra of applications in medicine, pharmaceuticals, cosmetics, and agriculture. Particular attention is paid to drimane sesquiterpenoids that exhibit certain biological properties, especially those with anticancer, antimicrobial, antifungal, antimalarial, antidiabetic, etc., activities [3,4,5,6,7,8,9].

On the other hand, many pharmaceuticals that mimic bioactive natural products are known to contain heterocycles. Among them are the benzimidazole derivatives that have found practical applications in various fields because they have numerous pharmacological activities such as antihypertensive, anticancer, antiviral, antidiabetic, antimicrobial, etc. [10].

The synthesis of molecules with a hybrid skeleton is often used in the design of drugs and especially in preparations with a promising biological activity. This approach is based on the combination of several pharmacophores which produce compounds with a combined skeleton and have a higher bioactivity than known drugs.

The synthesis of terpene—heterocyclic compounds has seen a vertiginous development in the last 10 years. A great number of molecular hybrids containing a terpene unit and one of the following heterocycles: diazine [11,12], 1,2,4-triazole and carbazole [13,14], azaheterocyclic [15,16], hydrazinecarbothioamide and 1,2,4-triazole [17], 1,3,4-oxadiazole and 1,3,4-thiadiazole [18], thiosemicarbazone and 1,3-thiazole [19], or benzothiazole [20], many of which showed excellent antifungal and/or antibacterial activity, were reported elsewhere.

The purpose of the present research was the development of original methods for the preparation of new terpene—heterocyclic derivatives based on the available natural diterpenoid—sclareol, and the designing of natural chiral molecules of interest to the pharmaceutical industry. Herein, we report the results of the synthesis of novel homodrimane sesquiterpenoids containing 2-substituted 1,3-benzimidazole and *N*-substituted 2-amino-1,3-benzimidazole, and their antimicrobial properties evaluation.

## 2. Results and Discussion

### 2.1. Synthesis and Characterization

Starting from sclareolide (**1**), carboxylic acids **2**, **5**, and **8** were obtained in five, six, and three steps, with overall yields of 81%, 62%, and 89%, respectively [13,21,22] (Figure 1). The intermediate carboxylic acid **12** was obtained based on (-)-sclareol **11** in two steps with the overall yield of 75% [23].

In continuation of our previous work, a series of new *N*-homodrimenoyl-2-amino-1,3-benzimidazoles was prepared starting from the intermediate carboxylic acids **2**, **5**, **8**, and **12**, via their acyl chlorides **3**, **6**, **9**, and **13** generated in situ. The desired *N*-substituted 2-amino-1,3-aminobenzimidazoles **4**, **7**, **10**, and **14** were obtained, with yields between 66–85%, by acylation of 2-amino-1,3-aminobenzimidazole with the mentioned homodrimane acyl chlorides under the mentioned conditions [12] (Figure 1).

According to the NMR spectra, the hybrids involved both heterocyclic and terpene units, and their accurate masses were confirmed by the high-resolution mass spectrometry (HRMS). All proton spectra of compounds **4**, **7**, **10**, and **14** include the signals of aromatic protons in a range of 7.09–7.48 ppm, together with the signals specific for terpene units such as singlets of C_8_-bonded methyl groups at 1.54–1.69 ppm and doublets of C_6_- and C_7_-bonded protons at 3.88 and 5.95, or C_7_-bonded methoxy groups at 3.52 ppm, and broad singlets of amine protons in a range of 7.00–7.29 ppm. The structures of the reported *N*-substituted 2-amino-benzimidazoles were additionally confirmed by the ^13^C NMR spectra.

Several attempts to obtain desired benzimidazoles by the direct heterocyclization of acids **2**, **5**, **8**, and **12** with *o*-phenylenediamine in the presence of 4N HCl [24], glacial AcOH [25] or BF_3_•OEt_2_ [26] gave no results. In the case of the treatment with triphenylphosphine and triethylamine [27], the monoacylated derivatives **15**, **17**, **19**, and **20** were afforded in the yields depicted in Figure 2. In the case of acids **2** and **5**, diacylated derivatives **16** and **18** were also obtained, respectively.

The structures of the synthesized compounds were confirmed by the ^1^H, ^13^C, ^15^N, and 2D NMR spectroscopy and by the HRMS analysis, and finally, in the case of amide **20**, by the single-crystal X-ray diffraction (XRD). The formation of compounds **15**, **17**, **19**, and **20** was proven, first of all, by the presence of signals attributed to aromatic protons from a a common phenylene unit in a range of 6.74–7.29 ppm, and broad singlets of aminic and amidic protons in a range of 3.79–3.88 ppm and 7.55–8.31 ppm, respectively. In addition, some individual signals, such as a singlet corresponding to protons of C_7_-bonded methoxy group at 3.36 ppm, a singlet corresponding to protons of C_8_-bonded acethoxy group at 1.92 ppm, or a doublet of doublets of C_6_- and C_7_-bonded protons at 5.90 and 5.95 ppm, confirmed the presence of a terpene unit. Those structures were fully confirmed by the carbon spectral data.

The structural analysis of compound **16** by the ^1^H, ^13^C, ^1^H/^1^H COSY, and the ^1^H/^13^C HSQC NMR spectra suggested the presence of an isolated spin system: CH_2_CH_2_CH_2_ (C_1_ to C_3_) (Figure 1). In the ^1^H/^13^C HMBC spectrum, the correlations of H-C_5_,_5′_ with two sp^2^ hybridized carbons (C_6_,_6′_, *δ*_C_ 129.1 and C_7_,_7′_, *δ*_C_ 129.3) have confirmed the presence of the Δ^6,7^ double bond, which was also supported by the correlations of H3-C_17_,_17′_ with C_7_,_7′_.

As mentioned above, the chemical composition and the crystal structure of compound **20** was confirmed via XRD. As shown in Figure 2, the asymmetric part of the unit cell consists of one molecular unit, which corresponds to that supposed on the base of the NMR spectra. There is no co-crystallized solvate molecule in the crystal. The values of the bond distances and angles are summarized in Appendix A. The analysis of the crystal structure showed the presence of different fragments that are potential proton donors or proton acceptors, which creates premises for noncovalent intermolecular interactions. Therefore, the main structural motif is characterized as a 2D supramolecular layer assembled via the network of N**−**H∙∙∙O hydrogen bonding interactions, as shown in Figure 3.

In continuation, the cyclodehydration of the resulting monoacylamides **15**, **17**, **19,** and **20** with *p*-TsOH in toluene [28] was performed. In the case of monoacylamides **15** and **19,** 2-substituted benzimidazoles **21** and **22** were obtained (Figure 3). The formation of the double unsaturated benzimidazole **22** from amides **15** and **19** can be explained by the elimination of the C_7_-methoxy group from compound **19** under acidic conditions, followed by the proton abstraction from the C_5_ position and, as result, isomerization of the Δ^6–7^ double bond into Δ^5–6^. Under similar conditions, monoacylamides **17** and **20** gave the same benzimidazole **23** (Figure 3). The formation of the Δ^8–9^ benzimidazole **23** derivative from compound **20** is a result of the C_8_-acetoxy group elimination.

According to the NMR spectra, the hybrid compounds **21**–**23** contained both heterocyclic and terpene units, and their accurate masses were confirmed by the HRMS analysis. The formation of the mentioned compounds was revealed, first of all, with the presence of the signals attributed to aromatic protons from a common 2-substituted-benzimidazole unit in a range of 7.18–7.24 ppm. Together with the signals specific for a terpene unit, such as singlets of C_6_- and C_7_-bonded protons at 5.83 and 5.97 ppm for compound **21**, C_6_-bonded proton at 5.69 ppm for compound **22** and broad singlets of aminic protons at 8.79–9.00 ppm were obtained. The structures of the reported benzimidazoles were additionally confirmed by the ^13^C NMR spectra.

The NMR data of compound **22** have been assigned on the base of the 1D (^1^H, ^13^C, DEPT-135º) and 2D homo- (^1^H/^13^C HSQC, ^1^H/^13^C HMBC and ^1^H/^1^H COSY-45º) correlation spectra. An analysis of the ^1^H, ^13^C, ^1^H/^1^H COSY and ^1^H/^13^C HSQC NMR spectra suggested the presence of two isolated spin systems: CH_2_CH_2_CH_2_ (C_1_ to C_3_) and CHCH_2_ (C_6_ to C_7_) (Figure 4). The rearrangement of the double bond of compound **22** was established by a detailed analysis of its ^1^H/^13^C HMBC spectrum. Thus, the observed correlations of H3-C_18_ with two sp^2^ hybridized carbons (C_5_, *δ*_C_ 125.3 and C_6_, *δ*_C_ 120.1) were indicative of the Δ^5,6^ double bond localization. The position of a nitrogen atom was confirmed by the ^1^H/^15^N HMBC spectra and supported by the correlations of an H2-C_11_/N cross-peak (Figure 4).

### 2.2. Antimicrobial Activity

All synthesized compounds were subjected to preliminary screening for their in vitro antifungal and antibacterial activities [29] against pure cultures of fungal species *Aspergillus niger*, *Fusarium solani*, *Penicillium chrysogenum*, *Penicillium frequentans*, and *Alternaria alternata* and both Gram-positive *Bacillus* sp. and Gram-negative *Pseudomonas aeruginosa* bacteria strains. The obtained minimum inhibitory concentration (MIC) values revealed that compounds **7** and **20** possess the highest antifungal (MIC 0.064 and 0.05 µg/mL, respectively,) and antibacterial (MIC 0.5 and 0.032 µg/mL, respectively,) activities, followed by compound **4** (MIC 1.6 and 4.0 µg/mL, respectively), which is comparable with the standards activity (Table 1, entries 1 and 5). Compounds **10**, **14,** and **19** have showed a moderate antifungal activity at MIC in a range from 0.80 to 1.16 µg/mL, and an antibacterial activity at MIC in a range from 3.90 to 6.0 µg/mL, vs the same standard (Table 1, entries 2–4). Compounds **15**, **16**, **17**, **18**, **21**, **22,** and **23** were found to be biologically inactive.

Based on the biological test data, it can be concluded that the activity of the reported compounds is largely determined by the nature of the heterocyclic unit. In this way, all *N*-homodrimenoyl-2-amino-1,3-benzimidazoles **4**, **7**, **10,** and **14** were found to be active while 2-homodrimenyl-1,3-benzimidazoles **21**–**23** were inactive. Probably, the activity of 2-amino-1,3-benzimidazoles is primarily due to the presence of an unsubstituted amine group. This is also confirmed by compounds **19** and **20** which were produced by condensation with *o*-phenylenediamine on a single amine group and proved to be active, unlike *bis*-amides **16** and **18** which were inactive. However, it can be mentioned that the activity of the hybrid compounds depends also on the terpene unit; more precisely, it depends on the combination of the functional groups from the B cycle. It can be assumed that the presence of the C_8_-acetate group and of the Δ^8–9^ double bond in the molecules of compounds **20** and **7**, in combination with the monosubstituted 2-amino-1,3-benzimidazolic or *o*-phenylenic fragment, make them the most active in this series. The presence of the Δ^8–9^ and Δ^6–7^ double bonds or the Δ^8–9^ bond and the C_7_-methoxy group in the molecules of compounds **4**, **10**, and **19** and of the C_8_-acetate group in compound **14**, in combination with the 2-amino-1,3-benzimidazolic fragment or *o*-phenylenic fragment, influences the activity to a lesser extent. Compounds **15** and **17**, which only contained a double bond Δ^8–9^ and Δ^6–7^ in combination with the monosubstituted *o*-phenylenic fragment, were inactive.

## 3. Materials and Methods

### 3.1. Synthesis and Characterization

The IR spectra were recorded on a Spectrum 100 FT-IR spectrometer (Perkin-Elmer, Shelton, CT, USA) using an ATR technique. The ^1^H, ^13^C, and ^15^N NMR (400, 100, and 40 MHz, respectively,) and COSY, ^1^H–^13^C HSQC, ^1^H–^13^C HMBC, DEPT, and ^1^H–^15^N HSQC, ^1^H–^15^N HMBC spectra were acquired on a Bruker Avance DRX 400 spectrometer (Bruker BioSpin, Rheinstetten, Germany) in CDCl_3_ (NMR spectra for all of the compounds are available online, see Appendix A). The ^1^H NMR chemical shifts were reported relative to the residual solvent protons as internal standards (7.26 ppm). The solvent carbon atoms served as internal standard for the ^13^C NMR spectra (77.0 ppm). The ^15^N NMR spectra were obtained using MeNO_2_ (380.5 ppm) and urea (73.4 ppm) as internal standards. Optical rotations measurements were performed on a Jasco DIP-370 polarimeter (Rudolph Research Analytical, Hackettstown, NJ, USA) with a 10 cm microcell. Melting points were determined on a Boetius (VEB Analytik, DDR) hot stage apparatus and were not uncorrected. The run of reactions and the purity of products were examined by TLC on Merck silica gel 60 plates, eluent CH_2_Cl_2_, or a mixture of CH_2_Cl_2_–MeOH, 99:1; 49:1. Visualization was achieved by the treatment with conc. H_2_SO_4_ and heating at 80 °C or using an UV lamp (254 or 365 nm). All solvents were purified and dried by standard techniques prior to use.

Compounds **4**, **7**, **10,** and **14** (General method).

The solution of one of the acids **2** (248 mg, 1 mmol), **5** (250 mg, 1 mmol), **8** (280 mg, 1 mmol), or **12** (310 mg, 1 mmol) dissolved in anhydrous C_6_H_6_ (5 mL) was treated with a solution of COCl_2_ (0.95 mL, 11 mmol) dissolved in C_6_H_6_ (2.5 mL). The reaction mixture was stirred at room temperature for 1 h and then refluxed for 1 h. The C_6_H_6_ and excess of COCl_2_ were removed at a reduced pressure on a rotary evaporator. Next, 2-aminobenzimidazole (225 mg, 1.5 mmol) was added to the solution of an acyl chloride **3**, **6**, **9,** or **13** in CH_2_Cl_2_ (10 mL), and the resulting mixtures were stirred at r.t. for 3 h, then refluxed for 4–6 h. After cooling, the precipitates were filtered off, washed with CH_2_Cl_2_, and the filtrates were concentrated to dryness at a reduced pressure on a rotary evaporator. The crude reaction products were purified by silica gel flash chromatography (1 → 3% MeOH/CH_2_Cl_2_).

1-(2-Amino-1*H*-benzo[d]imidazol-1-yl)-2-((8a*S*)-2,5,5,8a-tetramethyl-4a,5,6,7,8,8a-hexahydronaphthalen-1-yl)ethanone **4**. (239mg, 66%), colorless oil. ${\left[\alpha \right]}_{D}^{20}$ −45.9 (*c* 3.4, CHCl_3_). IR spectrum, ν, cm^−1^: 729, 906, 1111, 1165, 1262, 1334, 1461, 1598, 1643, 1708, 2925, 3436. ^1^H NMR (400 MHz, CDCl_3_) *δ* 0.91 (3H, s, 10-C*H_3_*), 0.98 (3H, s, 4-C*H_3_*), 0.98 (3H, s, 4-C*H_3_*), 1.13–1.60 (6H, m, 3CH_2_), 1.69 (3H, s, 8-C*H_3_*), 2.22 (1H, t, *J* = 2.6 Hz, H-5), 3.72 (1H, d, *J* = 17.2 Hz, H-11), 3.81 (1H, d, *J* = 17.8 Hz, H-11), 5.88 (1H, dd, *J*= 9.3, 2.5 Hz, H-6), 5.95 (1H, dd, *J*= 9.6, 3.0 Hz, H-7), 7.10 (1H, td, *J* = 8.0, 1.1 Hz, H-Ar), 7.14 (1H, br.s, NH), 7.26 (1H, t, *J* = 8.0 Hz, H-Ar), 7.37 (1H, t, *J* = 2.3 Hz, H-Ar), 7.47 (1H, d, *J* = 8.2 Hz, H-Ar). ^13^C NMR (100 MHz, CDCl_3_) *δ* 15.3 (C-20), 18.3 (C-17), 18.8 (C-2), 22.7 (C-18), 32.3 (C-19), 33.0 (C-4), 34.7 (C-1), 36.9 (C-11), 38.4 (C-10), 40.7 (C-3), 52.3 (C-5), 113.2 (Ar), 117.0 (Ar), 120.6 (Ar), 125.0 (Ar), 128.7 (C-6), 129.0 (C-7), 129.6 (C-8), 129.7 (Ar), 134.5 (C-9), 142.8 (Ar), 155.0 (C=N), 173.1 (C-12). ^15^N NMR (40 MHz, CDCl_3_) *δ* 62, 196. HRMS (ESI) calculated for C_23_H_29_N_3_O [M + H]^+^, 363.23106. Found: 363.24188.

1-(2-Amino-1*H*-benzo[d]imidazol-1-yl)-2-((8a*S*)-2,5,5,8a-tetramethyl-3,4,4a,5,6,7,8,8a-octahydronaphthalen-1-yl) ethanone **7**. (310 mg, 85%), mp 91–92 °C, ${\left[\alpha \right]}_{D}^{20}$ 67.9 (*c* 2.7, CHCl_3_). IR spectrum, ν, cm^−1^: 688, 719, 753, 894, 1110, 1164, 1262, 1336, 1461, 1540, 1599, 1656, 1709, 2922, 3431. ^1^H NMR (400 MHz, CDCl_3_) *δ* 0.85 (3H, s, 4-C*H_3_*), 0.92 (3H, s, 4-C*H_3_*), 1.01 (3H, s, 10-C*H_3_*), 1.17–1.23 (2H, m, CH_2_), 1.33 (1H, dd, *J* = 12.5, 1.3 Hz, H-5), 1.38–1.50 (4H, m, 2CH_2_), 1.54 (3H, s, 8-C*H_3_*), 1.70–1.78 (2H, m, CH_2_), 2.10 (1H, dd, *J* = 18.2, 5.6 Hz, H-7), 2.20–2.29 (1H, m, H-7), 3.67 (1H, d, *J* = 12.2, H-11), 3.72 (1H, d, *J* = 17.8, H-11), 7.01 (2H, br.s, NH_2_), 7.11 (1H, t, *J* = 8.1 Hz, H-Ar), 7.26 (1H, t, *J* = 7.5 Hz, H-Ar), 7.37 (1H, d, *J* = 7.7 Hz, H-Ar), 7.48 (1H, d, *J* = 8.2 Hz, H-Ar). ^13^C NMR (100 MHz, CDCl_3_) *δ* 18.8 (C-2), 18.9 (C-6), 20.0 (C-17), 21.6 (C-18), 33.1 (C-19), 33.3 (C-4), 33.5 (C-7), 36.0 (C-1), 37.4 (C-11), 38.4 (C-10), 41.4 (C-3), 51.4 (C-5), 113.3, 117.0, 120.6), 124.7, 129.8, 142.9 (Ar), 131.7 (C-8), 132.4 (C-9), 154.9 (C=N), 173.3 (C-12). ^15^N NMR (40 MHz, CDCl_3_) *δ* 60, 191. HRMS (ESI) calculated for C_23_H_31_N_3_O [M + H]^+^, 365.24671. Found: 365.25388.

1-(2-Amino-1*H*-benzo[d]imidazol-1-yl)-2-((3R,8a*S*)-3-methoxy-2,5,5,8a-tetramethyl-3,4,4a,5,6,7,8,8a-octahydronaphthalen-1-yl)ethanone **10**. (304 mg, 77%), mp 105–106 °C, ${\left[\alpha \right]}_{D}^{20}$ 75.6 (*c* 1.4, CHCl_3_). IR spectrum, ν, cm^−1^: 739, 755, 1075, 1166, 1262, 1341, 1460, 1542, 1595, 1647, 1707, 2926, 3434. ^1^H NMR (400 MHz, CDCl_3_) *δ* 0.87 (3H, s, 10-C*H_3_*), 0.94 (3H, s, 4-C*H_3_*), 0.96 (3H, s, 4-C*H_3_*), 1.16–1.27 (2H, m, CH_2_), 1.39–1.52 (4H, m, 2CH_2_), 1.55 (1H, t, *J* = 3.4 Hz, H-5), 1.59 (1H, d, *J* = 1.6 Hz, H-6), 2.04 (1H, d, *J* = 2.0, H-6), 1.63 (3H, s, 8-C*H_3_*), 3.40 (3H, s, 7-OC*H_3_*), 3.38 (1H, d, *J* = 6.9 Hz, H-11), 3.52 (2H, d, *J* = 5.7 Hz, H-11 and H-7), 7.00 (2H, br.s, NH_2_), 7.09 (1H, dt, *J* = 7.7, 1.2 Hz, H-Ar), 7.25 (1H, dt, *J* = 7.9, 0.7 Hz, H-Ar), 7.37 (1H, dd, *J* = 7.9, 0.7 Hz, H-Ar), 7.43 (1H, d, *J* = 8.2 Hz, H-Ar). ^13^C NMR (100 MHz, CDCl_3_) *δ* 18.1 (C-17), 18.7 (C-2), 22.6 (C-6), 21.6 (C-18), 32.8 (C-19), 32.9 (C-4), 35.6 (C-1), 37.2 (C-11), 39.2 (C-10), 41.1 (C-3), 45.7 (7-OCH_3_), 56.8 (C-5), 78.8 (C-7), 113.1, 116.9, 120.6, 124.9, 129.5, 142.6 (Ar), 131.8 (C-8), 137.2 (C-9), 154.9 (C=N), 171.9 (C-12). ^15^N NMR (40 MHz, CDCl_3_) *δ* 65, 195. HRMS (ESI) calculated for C_24_H_33_N_3_O_2_ [M + H]^+^, 395.25728. Found: 395.26447.

(1*R*,2*R*,8a*S*)-1-(2-(2-amino-1*H*-benzo[d]imidazol-1-yl)-2-oxoethyl)-2,5,5,8a-tetramethyldecahydronaphthalen-2-yl acetate **14**. (360 mg, 85%), colorless oil. ${\left[\alpha \right]}_{D}^{20}$ −2.9 (*c* 7.2, CHCl_3_). IR spectrum, ν, cm^−1^: 728, 907, 1113, 1164, 1246, 1306, 1382, 1461, 1598, 1642, 1713, 2929, 3439. ^1^H NMR (400 MHz, CDCl_3_) *δ* 0.79 (3H, s, 10-C*H_3_*), 0.89 (3H, s, 4-C*H_3_*), 0.92 (3H, s, 4-C*H_3_*), 1.09–1.22 (3H, m, H-5, CH_2_), 1.29–1.44 (4H, m, 2CH_2_), 1.56 (3H, s, 8-C*H_3_*), 1.66–1.76 (2H, m, CH_2_), 1.78 (3H, s, 8-OCOC*H_3_*), 1.85 (1H, s, H-9), 2.73–2.80 (2H, m, H-7), 2.93 (1H, dd, *J* = 17.2, 4.8 Hz, H-11), 3.11 (1H, dd, *J* = 17.4, 4.3 Hz, H-11), 7.29 (2H, br.s, NH_2_), 7.10 (1H, td, *J* = 7.5, *J* = 1.4 Hz, H-Ar), 7.24 (1H, td, *J* = 7.6, 0.6 Hz, H-Ar), 7.34 (1H, d, *J* = 8.0 Hz, H-Ar), 7.45 (1H, d, *J* = 8.0 Hz, H-Ar). ^13^C NMR (100 MHz, CDCl_3_) *δ* 16.2 (C-17), 18.1 (C-2), 19.9 (C-6), 20.8 (C-18), 21.3 (8-OCOCH_3_), 33.1 (C-4), 33.2 (C-19), 38.7 (C-7), 38.8 (C-1), 34.5 (C-11), 38.7 (C-10), 41.6 (C-3), 53.3 (C-5), 55.3 (C-9), 86.2 (C-8), 113.1, 116.8, 120.7, 124.9, 129.6, 142.5 (Ar), 155.1 (C=N), 169.9 (8-OCOCH_3_), 174.6 (C-12). ^15^N NMR (40 MHz, CDCl_3_) *δ* 60, 190. HRMS (ESI) calculated for C_25_H_35_N_3_O_3_ [M + H]^+^, 425.26784. Found: 425.27448.

Compounds **15**, **17**, **19** and **20** (General method).

One of the acids **2** (248 mg, 1 mmol), **5** (250 mg, 1 mmol), **8** (280 mg, 1 mmol), or **12** (310 mg, 1 mmol) was added to an ice bath-cooled solution of Ph_3_P (786 mg, 3 mmol) and Et_3_N (0.16 mL, 1.2 mmol) dissolved in anhydrous CCl_4_ (7 mL). After 10 min of stirring, the solution of *o*-phenylendiamine (150 mg, 1.2 mmol) dissolved in anhydrous CCl_4_ (3 mL) was added, and the reaction mixture was refluxed under stirring for 6 h. The solvents were removed under a reduced pressure on a rotary evaporator to dryness, and the crude reaction products were subjected to silica gel flash column chromatography (CH_2_Cl_2_ →2% MeOH/CH_2_Cl_2_).

*N*-(2-aminophenyl)-2-((8a*S*)-2,5,5,8a-tetramethyl-4a,5,6,7,8,8a-hexahydronaphthalen-1-yl)acetamide **15**. (182 mg, 57%), yellow oil. ${\left[\alpha \right]}_{D}^{20}$ 0.79 (*c* 0.7, CHCl_3_). IR spectrum, ν, cm^−1^: 730, 907, 1033, 1370, 1455, 1503, 1591, 1654, 2926, 3268, 3354. ^1^H NMR (400 MHz, CDCl_3_) *δ* 0.88 (3H, s, 10-C*H_3_*), 0.97 (3H, s, 4-C*H_3_*), 0.99 (3H, s, 4-C*H_3_*), 1.12–1.64 (6H, m, 3CH_2_), 1.86 (3H, s, 8-C*H_3_*), 2.07 (1H, t, *J* = 2.5 Hz, H-5), 3.12 (1H, d, *J* = 17.2 Hz, H-11), 3.34 (1H, d, *J* = 17.2 Hz, H-11), 3.83 (2H, br.s,NH_2_), 5.90 (1H, dd, *J* = 9.5, 2.3 Hz, H-6), 5.95 (1H, dd, *J* = 9.6, 2.7 Hz, H-7), 6.78–6.82 (2H, m, H-Ar), 7.06 (1H, td, *J* = 7.6, 1.4 Hz, H-Ar), 7.14 (1H, dd, *J* = 8.1, 1.4 Hz, H-Ar), 7.67 (1H, br.s, NH). ^13^C NMR (100 MHz, CDCl_3_) *δ* 15.2 (C-20), 18.4 (C-17), 18.8 (C-2), 22.7 (C-18), 32.4 (C-19), 33.1 (C-4), 35.0 (C-1), 35.9 (C-11), 39.2 (C-10), 40.9 (C-3), 53.6 (C-5), 118.2, 119.4, 124.5, 124.7, 127.1, 140.8 (Ar), 128.9 (C-6), 129.1 (C-7), 130.0 (C-8), 138.2 (C-9), 169.8 (C-12). ^15^N NMR (40 MHz, CDCl_3_) *δ* 51, 126. HRMS (ESI) calculated for C_22_H_30_N_2_O [M + H]^+^, 338.23581. Found: 338.24301.

(*S*)-*N*,*N*′-(1,2-phenylene)bis(2-((8aS)-2,5,5,8a-tetramethyl-4a,5,6,7,8,8a-hexahydrona-phthalen-1-yl)acetamide) **16**. (34 mg, 12%), yellow oil. ${\left[\alpha \right]}_{D}^{20}$ −140.16 (*c* 1.7, CHCl_3_). IR spectrum, ν, cm^−1^: 728, 907, 1369, 1445, 1511, 1599, 1662, 2926, 3264. ^1^H NMR (400 MHz, CDCl_3_) *δ* 0.86 (6H, s, 10-C*H_3_* and 10ʹ-C*H_3_*), 0.97 (6H, s, 4-C*H_3_* and 4ʹ-C*H_3_*), 0.98 (6H, s, 4-C*H_3_* and 4ʹ-C*H_3_*ʹ), 1.09–1.62 (12H, m, 3CH_2_), 1.80 (6H, s, 8-C*H_3_* and 8ʹ-C*H_3_*), 2.18 (2H, t, *J* = 2.5, H-5 and H-5ʹ), 3.08 (2H, d, *J* = 16.9 Hz, H-11 and H-11ʹ), 3.25 (2H, d, *J* = 16.9 Hz, H-11 and H-11ʹ), 5.90 (2H, dd, *J* = 9.4, 2.1 Hz, H-6 and H-6ʹ), 5.94 (2H, dd, *J* = 9.7, 2.4 Hz, H-7 and H-7ʹ), 7.19 (4H, dd, *J* = 5.7, 3.4 Hz, H-Ar), 7.44 (4H, dd, *J* = 7.4, 3.4 Hz, H-Ar), 8.12 (2H, s, 2NH). ^13^C NMR (100 MHz, CDCl_3_) *δ* 15.2 (C-20 and C-20ʹ), 18.5 (C-2 and C-2ʹ), 18.8 (C-17 and C-17ʹ), 22.7 (C-18 and C-18ʹ), 32.4 (C-19 and C-19ʹ), 33.4 (C-4 and C-4ʹ), 35.0 (C-1 and C-1ʹ), 36.1 (C-11 and C-11ʹ), 39.2 (C-10 and C-10ʹ), 40.8 (C-3 and C-3ʹ), 52.9 (C-5 and C-5ʹ), 124.8, 126.2, 129.9 (Ar), 128.9 (C-6 and C-6ʹ), 129.1 (C-7 and C-7ʹ), 130.6 (C-8 and C-8ʹ), 137.5 (C-9 and C-9ʹ), 170.7 (C-12 and C-12ʹ). ^15^N NMR (40 MHz, CDCl_3_) *δ* 123. HRMS (ESI) calculated for C_38_H_52_N_2_O_2_ [M + H]^–^, 568.40288. Found: 568.39551.

*N*-(2-aminophenyl)-2-((8a*S*)-2,5,5,8a-tetramethyl-3,4,4a,5,6,7,8,8a-octahydronaphtha-len-1-yl)acetamide **17**. (25 mg, 8%), yellow oil. ${\left[\alpha \right]}_{D}^{20}$ 99.88 (*c* 0.4, CHCl_3_). IR spectrum, ν, cm^−1^: 738, 756, 1018, 1086, 1141, 1274, 1317, 1348, 1462, 1478, 1625, 1721, 2918, 3215. ^1^H NMR (400 MHz, CDCl_3_) *δ* 0.85 (3H, s, 4-C*H_3_*), 0.91 (3H, s, 4-C*H_3_*), 1.00 (3H, s, 10-C*H_3_*), 1.17–1.51 (7H, m, 3CH_2_, H-5), 1.55 (3H, s, 8-C*H_3_*), 1.61–1.75 (2H, m, CH_2_), 3.79 (2H, br.s. NH_2_), 2.04–2.29 (2H, m, H-7) 3.74 (1H, d, *J* = 19.1 Hz, H-11), 4.02 (1H, d, *J* = 19.1 Hz, H-11), 7.04 (1H, dd, *J* = 7.6, 1.1 Hz, H-Ar), 7.13 (1H, td, *J* = 7.8, 1.4 Hz, H-Ar), 7.19 (1H, td, *J* = 7.6, 1.3 Hz, H-Ar), 8.21 (1H, d, *J* = 7.4 Hz, H-Ar), 8.31 (1H, s, NH). ^13^C NMR (100 MHz, CDCl_3_) *δ* 18.9 (C-2), 18.9 (C-6), 19.8 (C-20), 20.1 (C-17), 21.7 (C-18), 33.1 (C-19), 33.3 (C-4), 33.7 (C-7), 36.0 (C-1), 36.2 (C-11), 38.4 (C-10), 41.5 (C-3), 51.4 (C-5), 108.8, 116.2, 122.8, 124.5, 127.6, 127.8 (Arʹ), 130.6 (C-8), 133.4 (C-9), 172.4 (C-12). ^15^N NMR (40 MHz, CDCl_3_) *δ* 52, 126. HRMS (ESI) calculated for C_22_H_32_N_2_O [M + H]^–^, 340.25146. Found: 340.19998.

(*S*)-*N*,*N*′-(1,2-phenylene)bis(2-((8a*S*)-2,5,5,8a-tetramethyl-3,4,4a,5,6,7,8,8a-octahydro-naphthalen-1-yl)acetamide) **18**. (60 mg, 21%), yellow oil. ${\left[\alpha \right]}_{D}^{20}$ 83.62 (*c* 3.6, CHCl_3_). IR spectrum, ν, cm^−1^: 751, 1043, 1161, 1295, 1376, 1443, 1511, 1599, 1663, 2865, 2923, 3264. ^1^H NMR (400 MHz, CDCl_3_) *δ* 0.86 (6H, s, 4-C*H_3_* and 4ʹ-C*H_3_*), 0.93 (6H, s, 4-C*H_3_* and 4ʹ-C*H_3_*), 1.00 (6H, s, 10-C*H_3_* and 10ʹ-C*H_3_*), 1.16–1.62 (14H, m, 6CH_2_, H-5 and H-5ʹ), 1.67 (6H, s, 8-C*H_3_* and 8ʹ-C*H_3_*), 1.70–1.81 (4H, m, 2CH_2_), 2.13–2.23 (4H, m, H-7 and H-7ʹ), 3.09 (2H, d, *J* = 17.6 Hz, H-11 and H-11ʹ), 3.23 (2H, d, *J* = 17.96 Hz, H-11 and H-11ʹ), 7.11 (4H, dd, *J* = 5.9, 3.5 Hz, H-Ar), 7.36 (4H, dd, *J* = 5.9, 3.2 Hz, H-Ar), 8.11 (2H, s, 2NH). ^13^C NMR (100 MHz, CDCl_3_) *δ* 18.9 (C-2 and C-2ʹ), 18.9 (C-6 and C-6ʹ), 20.1 (C-20 and C-20ʹ), 20.3 (C-17 and C-17ʹ), 21.7 (C-18 and C-18ʹ), 33.2 (C-19 and C-19ʹ), 33.4 (C-4 and C-4ʹ), 33.6 (C-7 and C-7ʹ), 36.3 (C-11 and C-11ʹ), 36.7 (C-1 and C-1ʹ), 38.9 (C-10 and C-10ʹ), 41.5 (C-3 and C-3ʹ), 51.8 (C-5 and C-5ʹ), 125.1, 126.1, 130.7 (Ar), 132.2 (C-8 and C-8ʹ), 135.4 (C-9 and C-9ʹ), 171.1 (C-12 and C-12ʹ). ^15^N NMR (40 MHz, CDCl_3_) *δ* 125. HRMS (ESI) calculated for C_38_H_56_N_2_O_2_ [M + H]^–^, 572.43418. Found: 572.42700.

*N*-(2-aminophenyl)-2-((3*R*,8a*S*)-3-methoxy-2,5,5,8a-tetramethyl-3,4,4a,5,6,7,8,8a-octa-hydronaphthalen-1-yl)acetamide **19**. (204 mg, 58%), yellow oil. ${\left[\alpha \right]}_{D}^{20}$ 84.78 (*c* 7.9, CHCl_3_). IR spectrum, ν, cm^−1^: 742, 1075, 1341, 1456, 1516, 1656, 2926, 3260, 3358. ^1^H NMR (400 MHz, CDCl_3_) *δ* 0.87 (3H, s, 10-C*H_3_*), 0.93 (3H, s, 4-C*H_3_*), 0.97 (3H, s, 4-C*H_3_*), 1.16–1.27 (3H, m, CH_2_), 1.41–1.63 (5H, m, H-5, 2CH_2_), 21.82 (3H, s, 8-C*H_3_*), 0.03 (1H, d, *J* = 14.2 Hz, CH_2_), 3.12 (1H, d, *J* = 17.8 Hz) and 3.25 (1H, d, *J* = 17.6 Hz, H-11), 3.37 (3H, s, 7-OCH_3_), 3.48 (1H, d, *J* = 3.6 Hz, H-7), 3.79 (2H,br.s. NH_2_), 6.74–6.79 (2H, m, H-Ar), 7.01 (1H, dt, *J* = 7.6, 1.2 Hz, H-Ar), 7.29 (1H, dd, *J* = 7.7, 1.1 Hz, H-Ar), 7.62 (1H, s, NH). ^13^C NMR (100 MHz, CDCl_3_) *δ* 18.4 (C-20), 18.5 (C-17), 18.7 (C-2), 21.6 (C-18), 22.4 (C-6), 32.8 (C-19), 32.9 (C-4), 35.5 (C-1), 36.6 (C-11), 39.7 (C-10), 41.2 (C-3), 45.9 (7-OCH_3_), 56.8 (C-5), 79.0 (C-7), 117.7, 119.2, 124.3, 126.6, 140.9 (Ar), 132.1 (C-8), 140.2 (C-9), 169.1 (C-12). ^15^N NMR (40 MHz, CDCl_3_) *δ* 51, 121. HRMS (ESI) calculated for C_23_H_34_N_2_O_2_ [M + H]^–^, 370.26230. Found: 370.25464.

(1*R*,2*R*,8a*S*)-1-(2-((2-aminophenyl)amino)-2-oxoethyl)-2,5,5,8a-tetramethyldecahydro-naphthalen-2-yl acetate **20**. (187 mg, 49%), mp 188–189 °C, ${\left[\alpha \right]}_{D}^{20}$ 11.09 (*c* 2.4, CHCl_3_). IR spectrum, ν, cm^−1^: 746, 1025, 1127, 1248, 1366, 1459, 1526, 1654, 1720, 2942, 3261, 3675. ^1^H NMR (400 MHz, CDCl_3_) *δ* 0.79 (3H, s, 10-C*H_3_*), 0.86 (6H, s, 4-C*H_3_* and 4-C*H_3_*), 1.07–1.39 (5H, m, H-5, 2CH_2_), 1.41–1.44 (1H, m, CH_2_), 1.51 (3H, s, 8-C*H_3_*), 1,55–1.76 (4H, m, 2CH_2_), 1.92 (3H, s, 8-OCOC*H_3_*), 2.34 (H, s, H-9), 2.36 (1H, dd, *J* = 18.3, 6.7 Hz, H-11), 2.53 (1H, dd, *J* = 18.3, 6.8 Hz, H-11), 2.71 (1H, dt, *J* = 12.4, 3.1 Hz, H-7), 3.88 (2H, br.s., NH_2_), 6.77 (2H, t, *J* = 7.8 Hz, H-Ar), 7.03 (1H, td, *J* = 7.6, 1.1 H-Ar), 7.17 (1H, dd, *J* = 8.2, 1.3 Hz, H-Ar), 7.56 (1H, s, NH). ^13^C NMR (100 MHz, CDCl_3_) *δ* 15.6 (C-20), 18.2 (C-2), 19.9 (C-18 and C-6), 21.4 (8-OCOCH_3_), 23.0 (C-17), 33.1 (C-4), 33.2 (C-19), 33.6 (C-11), 38.7 (C-7), 39.2 (C-1 and C-10), 41.6 (C-3), 55.5 (C-5), 56.0 (C-9), 87.4 (C-8), 118.2, 119.4, 124.6, 124.8, 126.8, 140.5 (Ar), 170.3 (C-12), 172.5 (8-OCOCH_3_). ^15^N NMR (40 MHz, CDCl_3_) *δ* 53, 125. HRMS (ESI) calculated for C_24_H_36_N_2_O_3_ [M + H]^–^, 400.27259. Found: 400.19977.

Compounds **21**–**23** (General method).

A solution of monoacylated compounds **15** (168 mg, 0.5 mmol), **17** (170 mg, 0.5 mmol), **19** (185 mg, 0.5 mmol), or **20** (200 mg, 0.5 mmol) and *para*-toluenesulfonic acid (187 mg, 1 mmol) in toluene (3.5 mL) was stirred for 24 h at reflux. Then the solvent was evaporated, and the residue was diluted with CH_2_Cl_2_ (20 mL), washed with aqueuous NaHCO_3_ 5%, dried over Na_2_SO_4_, and concentrated. The crude reaction products were purified by silica gel flash chromatography (CH_2_Cl_2_).

2-(((8a*S*)-2,5,5,8a-tetramethyl-4a,5,6,7,8,8a-hexahydronaphthalen-1-yl)methyl)-1*H*-benzo[d]imidazole **21**. (51 mg, 32% or 44 mg, 28%), colorless oil. ${\left[\alpha \right]}_{D}^{20}$ 6.70 (*c* 0.2, CHCl_3_). IR spectrum, ν, cm^−1^: 742, 1016, 1117, 1272, 1364, 1420, 1453, 1524, 1590, 1622, 1719, 2925, 3054. ^1^H NMR (400 MHz, CDCl_3_) *δ* 0.78 (6H, s, 4-C*H_3_* and 4-C*H_3_*), 0.86 (3H, s, 10-C*H_3_*), 0.93–1.16 (4H, m, 2CH_2_), 1.52–1.62 (2H, m, CH_2_), 1.93 (3H, s, 8-C*H_3_*), 2.35 (1H, t, *J* = 3.4 Hz, H-5), 3.83 (1H, d, *J* = 16.7 Hz, H-11), 3.94 (1H, d, *J* = 15.6 Hz, H-11), 5.83 (1H, dd, *J* = 9.4, 6.1 Hz, H-6), 5.97 (1H, d, *J* = 9.6 Hz, H-7), 7.18–7.24 (4H, m, H-Ar), 8.79 (1H, br.s, NH). ^13^C NMR (100 MHz, CDCl_3_) *δ* 18.8 (C-20), 20.1 (C-2), 20.4 (C-17), 20.4 (C-18), 28.2 (C-11), 30.7 (C-19), 34.4 (C-4), 35.8 (C-1), 37.5 (C-10), 41.3 (C-3), 52.7 (C-5), 117.5, 122.1, 125.3, 127.6, 129.0, 132.1 (Ar), 128.2 (C-6), 129.2 (C-7), 129.2 (C-8), 134.3 (C-9), 153.5 (C=N, benzimidazole). ^15^N NMR (40 MHz, CDCl_3_) *δ* 130, 247. HRMS (ESI) calculated for C_22_H_28_N_2_ [M + H]^+^, 320.22525. Found: 320.23145.

(*R*)-2-((2,5,5,8a-tetramethyl-3,5,6,7,8,8a-hexahydronaphthalen-1-yl)methyl)-1*H*-benzo[d]imidazole **22**. (27 mg, 17% or 22 mg, 14%), mp 188–189 °C, ${\left[\alpha \right]}_{D}^{20}$ −138.9 (*c* 1.0, CHCl_3_). IR spectrum, ν, cm^−1^: 741, 1024, 1270, 1371, 1417, 1453, 1522, 1590, 1622, 2925, 3051. ^1^H NMR (400 MHz, CDCl_3_) *δ* 0.84 (3H, s, 4-C*H_3_*), 0.90 (3H, s, 4-C*H_3_*), 1.00 (3H, s, 10-C*H_3_*), 1.10–1.16 (1H, m, CH_2_), 1.55–1.79 (4H, m, 2CH_2_), 1.85 (3H, s, 8-C*H_3_*), 2.07–2.31 (3H, m, CH_2_), 3.90 (1H, d, *J* = 17.2 Hz, H-11), 4.04 (1H, d, *J* = 17.2 Hz, H-11), 5.69 (1H, t, *J* = 3.9 Hz, H-6), 7.19–7.23 (4H, m, H-Ar), 8.80 (1H, br.s, NH). ^13^C NMR (100 MHz, CDCl_3_) *δ* 20.7 (C-20), 20.8 (C-17), 23.0 (C-18), 23.3 (C-2), 24.5 (C-19), 27.5 (C-1), 29.1 (C-11), 29.9 (C-7), 32.4 (C-3), 34.5 (C-4), 37.8 (C-10), 125.3 (C-5), 117.1, 120.1, 121.9, 128.2, 129.0 (Ar), 120.1 (C-6), 133.6 (C-8), 139.8 (C-9), 154.2 (C=N, benzimidazole). ^15^N NMR (40 MHz, CDCl_3_) *δ* 134, 243. HRMS (ESI) calculated for C_22_H_28_N_2_ [M + H]^+^, 320.22525. Found: 320.23151.

2-(((8a*S*)-2,5,5,8a-tetramethyl-3,4,4a,5,6,7,8,8a-octahydronaphthalen-1-yl)methyl)-1*H*-benzo[d]imidazole **23**. (80 mg, 42% or 97 mg, 51%), mp 175–176 °C, ${\left[\alpha \right]}_{D}^{20}$ 69.29 (*c* 1.0, CHCl_3_). IR spectrum, ν, cm^−1^: 740, 1015, 1118, 1270, 1375, 1415, 1453, 1519, 1591, 1622, 2926, 3054. ^1^H NMR (400 MHz, CDCl_3_) *δ* 0.83 (3H, s, 4-C*H_3_*), 0.89 (3H, s, 4-C*H_3_*), 1.00 (3H, s, 10-C*H_3_*), 1.07–1.54 (7H, m, 3CH_2_, H-5), 1.68 (3H, s, 8-C*H_3_*), 1.73–1.77 (2H, m, CH_2_), 2.14–2.23 (2H, m, CH_2_), 3.70 (1H, d, *J* = 17.2 Hz, H-11), 3.77 (1H, d, *J* = 17.2 Hz, H-11), 7.16–7.22 (4H, m, H-Ar), 9.00 (1H, br.s, NH). ^13^C NMR (100 MHz, CDCl_3_) *δ* 18.7 (C-2), 18.9 (C-6), 19.9 (C-20), 20.3 (C-17), 21.6 (C-18), 28.1 (C-11), 33.2 (C-19), 33.3 (C-4), 33.6 (C-7), 36.6 (C-1), 39.0 (C-10), 41.5 (C-3), 52.1 (C-5), 118.9, 121.1, 122.0, 128.2, 129.1 (Ar), 130.7 (C-8), 136.5 (C-9), 154.5 (C=N, benzimidazole). ^15^N NMR (40 MHz, CDCl_3_) *δ* 139, 238. HRMS (ESI) calculated for C_22_H_30_N_2_ [M + H]^+^, 322.24090. Found: 322.25364.

### 3.2. X-ray Crystallography

Single-crystal XRD data were collected on an Oxford-Diffraction XCALIBUR Eos CCD diffractometer with graphite-monochromated Mo-Kα radiation. A single crystal was positioned at 46 mm from the detector, and 600 frames were measured each for 25 s over 1° scan width. The unit cell determination and data integration were carried out using the CrysAlisPro package from Oxford Diffraction [30]. A multiscan correction for absorption was applied. The structure was solved with the software SHELXT using the intrinsic phasing method and refined by the full-matrix least-squares method on *F*^2^ with SHELXL [31,32]. Olex2 was used as an interface to the SHELX programs [33]. Nonhydrogen atoms were refined anisotropically. Hydrogen atoms were added in idealized positions and refined using a riding model. The positional parameters of disordered atoms were refined in combination with PART and AFIXI restraints using an anisotropic model for non-H atoms. In the absence of a significant anomalous scattering, the absolute configuration of the structures could not be reliably determined, and therefore, Friedel pairs were merged and any references to the Flack parameter were removed. The molecular plots were obtained using the Olex2 program. Selected crystallographic data and structure refinement details for are provided in Appendix A.

### 3.3. Antifungal and Antibacterial Activity Assay

Pure cultures of the fungi *Aspergillus niger*, *Fusarium*, *Penicillium chrysogenum*, *Penicillium frequentans*, and *Alternaria alternata* and bacteria *Pseudomonas aeruginosa* and *Bacillus* sp. were used as obtained from the American Type Culture Collection (ATCC). Suspensions of microorganisms in DMSO were prepared according to the direct colony method and the serial dilution procedure. The final concentration of the stock inoculum was 1·10^–4^ μg/mL. Both antifungal and antibacterial activities assays were performed by applying a mixture of a microorganism suspension and a solution of the target compound in a ratio 1:1 to Petri dishes with a solid medium: Merck Sabouraud agar or agar-agar. The DMSO did not have any inhibitory effect on the tested organisms.

## 4. Conclusions

A series of seven novel *N*-homodrimenoyl-2-amino-1,3-benzimidazoles and 2-homodrimenyl-1,3-benzimidazoles, four intermediate monoacylamides, and two bis-acylamides were designed, synthesized, and assessed as antimicrobial agents. Six of them showed high to moderate antifungal and antibacterial activities compared to those of the reference drugs.

## Data Availability

Not applicable.

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
