# Peer review of "Synthesis and Antimicrobial Activity Evaluation of Homodrimane Sesquiterpenoids with a Benzimidazole Unit"

_molecules, 2023, doi:10.3390/molecules28030933_

Round 1

Reviewer 1 Report

The reviewed article by:

 Lidia Lungu , Svetlana Blaja , Caleria Cucicova , Alexandru Ciocarlan , Alic Barba , Veaceslav Kulcițki , Sergiu  Shova , Nicoleta Vornicu , Elisabeta-Irina Geana , Ionel I. Mangalagiu , and Aculina Aricu, titled

„Synthesis and Antimicrobial Activity Evaluation of Homodrimane Ses-quiterpenoids with Benzimidazole Unit” contains 34 references.

Authors present the result of the synthesis of homodrimane sesquiterpenoids  containing 2-substituted 1,3-benzimidazole and N-substituted 2-amino-1,3-benzimidazole and their antimicrobial properties evaluation.

It is an essential element in the development of heteroorganic chemistry.

An important and significant element of the work, is the study of the antibacterial activity of the sythtetized compounds: numers 4,7,10,14,19 and 20.

The second one (number 7) and the last one (number20) showed higher antifungal and antibacterial activity.

This is the goal of the paper.

Reaction of compouds 2,5,8 and 12 with 2-aminobenzimidazole resulting in good yields of of the corresponding N-substituted 2-amino-1,3-aminobenzimidazoles 4, 7, 10 and 14 proceed via generated „in situ” acid chlorides:3,6,9 and 13.

Question to the Authors:

Why was this step omitted in the reaction with o-phenylenediamine?

It is also  primary amine (diamine).However aromatic.

Please comment and/or explain above matter.

In the part 3.Material and Methods: before numbers of obtained compounds, full chemical  names in accordance with nomenclature should be added .

Author Response

Dear Reviewer!

Thank you very much for suggestion and corrections, for the time and effort invested.

We did all the required in the first revision changes.

Best wishes.

Sincerely,

Prof., Dr. Aculina Aricu

Reviewer 2 Report

In this manuscript, the authors report the synthesis of some new homodrimane sesquiterpenoids with benzimidazole unit and in vitro antimicrobial activity. The structures of the synthesized compounds have been fully confirmed, including by X-ray diffraction. Their biological activities were evaluated on five species of fungi. Compounds 7 and 20 showed higher antifungal (MIC=0.064 and 0.05 μg/mL) and antibacterial(MIC=0.05 and 0.032 μg/mL) activities compared to the standards caspofungin (MIC=0.32 μg/mL) and kanamycin (MIC=2.0 μg/mL). I recommend this manuscript be accepted for publication in this journal after minor revisions.

1. In the process of synthesis of monoacilated precursors 15, 17, 19, and 20 from carboxylic acids 2, 5, 8, 12 and o-phenylendiamine, what conditions have the author tried, such as reaction temperature, feed ratio, etc?

2. The letter “o” of o-phenylendiamine should be italicized in Scheme 2.

3. Supplementary Materials should be provided.

4. The format of references should be consistent. For example, references 1-10 are different from others.

Author Response

(The authors gave the same response as above.)
